



# Recurrence analysis of extreme event like data

Abhirup Banerjee[1,2], Bedartha Goswami[1,3], Yoshito Hirata[4], Deniz Eroglu[5], Bruno Merz[2,6],
Jürgen Kurths[1,7], and Norbert Marwan[1,8]

[1]Potsdam Institute for Climate Impact Research (PIK), Member of the Leibniz Association, 14412 Potsdam, Germany
[2]Institute of Environmental Science and Geography, University of Potsdam, 14476 Potsdam, Germany
[3]Currently at: University of Tübingen, 72074 Tübingen, Germany
[4]Faculty of Engineering, Information and Systems, University of Tsukuba, 1-1-1 Tennodai, Tsukuba, Ibaraki 305-8573, Japan
[5]Department of Bioinformatics and Genetics, Kadir Has University, 34083 Istanbul, Turkey
[6]Helmholtz Centre Potsdam, GFZ German Research Centre for Geosciences, Potsdam, Germany
[7]Institute of Physics, Humboldt Universität zu Berlin, Germany
[8]Institute of Geoscience, University of Potsdam, 14476 Potsdam, Germany

**Correspondence:** Abhirup Banerjee (banerjee@pik-potsdam.de)

**Abstract.** The identification of recurrences at various time scales in extreme event-like time series is challenging because of the rare occurrence of events which are separated by large temporal gaps. Most of the existing time series analysis techniques cannot be used to analyse extreme event-like time series in its unaltered form. The study of the system dynamics by reconstruction of the phase space using the standard delay embedding method is not directly applicable to event-like time series as

5  it assumes a Euclidean notion of distance between states in the phase space. The *edit distance* method is a novel approach that uses the point-process nature of events. We propose a modification of edit distance to analyze the dynamics of extreme event-like time series by incorporating a nonlinear function which takes into account the sparse distribution of extreme events and utilizes the physical significance of their temporal pattern. We apply the modified edit distance method to event-like data generated from point process as well as flood event series constructed from discharge data of the Mississippi River in USA, and

10  compute their recurrence plots. From the recurrence analysis, we are able to quantify the deterministic properties of extreme event-like data. We also show that there is a significant serial dependency in the flood time series by using the random shuffle surrogate method.

## 1 Introduction

15  One of the main challenges of the society is to understand and manage natural disasters, such as earthquakes, tsunamis, and floods, which often lead to big loss of economic assets and even lives. Flooding is an important example with a high societal relevance and affects more people globally than any other natural hazard. Globally, the expected annual costs of floods have been estimated to more than US$ 100 billion (Desai et al., 2015). Furthermore, climate change projections point to an increasing flood risk. Direct flood damages could rise by 160 to 240% and human losses by 70 to 83% in a 1.5°C warmer world (Dottori





et al., 2018). Climate change has already influenced river flood magnitudes (Bloschl et al., 2019), and has been related to increases in the intensity and frequency of heavy precipitation events aggregating flash flood and river flood risk (Donat et al., 2016; Kemter et al., 2020). Natural climate variability at different time scales may lead to flood-rich and flood-poor periods (Merz et al., 2018). In addition, human interventions in river systems and catchments also heavily influence flood magnitudes and frequencies (Hall et al., 2014). Since floods can massively affect life quality of our societies, it is desirable to understand the underlying dynamics and, thus, put forward precautionary measures to avert potential disasters.

The occurrence of extreme events is not random but rather a manifestation of complex dynamics, and such events tend to have long-term correlation. Comparing data of extreme events with other non-event data using standard methods (linear/nonlinear) is problematic, because the temporal sampling differs largely (time points of events vs. continuous sampling). Furthermore, linear methods such as the Fourier transform (Bloomfield, 2004) and wavelet analysis (Percival and Walden, 2007) are often insufficient to capture the full range of dynamics occurring due to the underlying nonlinearities (Marwan, 2019). Hence, defining a principled nonlinear method is necessary for the analysis of extreme event time series, in particular when the correlation or coupling between several variables is investigated.

Out of the various approaches to study nonlinear dynamical systems (Bradley and Kantz, 2015), the reconstruction of the phase space using delay coordinates (Takens, 1981) is a widely used method that allows us to estimate dynamical invariants by constructing a topologically equivalent dynamical trajectory of the original (often high-dimensional and unknown) dynamics from the measured (scalar) time series. In the delay coordinate approach, the distance between states in the phase space plays a pivotal role in describing the underlying dynamics of a system. After reconstructing the dynamical trajectory, we can extract further information about the dynamics of a system encoded in the evolution of the distances between the trajectories, e.g., through recurrence plots (Marwan et al., 2007), correlation dimension (Grassberger and Procaccia, 1983b), Kolmogorov entropy (Grassberger and Procaccia, 1983a), or Lyapunov exponents (Wolf et al., 1985).

Although there are powerful techniques based on phase space reconstruction of a wide range of nonlinear dynamical processes, they are not directly applicable to event-like time series. Extreme events such as flood, earthquakes, or solar flares are known to have long-term correlations (Jentsch et al., 2006). However, capturing the correlations of extreme events using such methods is difficult as the phase space reconstruction and the Euclidean distance for measuring the distances of states are not suitable for event-like time series because, by definition, extreme events are small in number and are separated by large temporal gaps. In this case, it becomes necessary to define an appropriate distance measure that can help analyze the dynamics of extreme event-like time series.

Event-like time series can be analyzed in their unaltered form by considering a time series of discrete events as being generated by a point process. Victor and Purpura (1997) presented a new distance metric called *edit distance* to calculate a distance between two spike trains (binary event sequences) as a measure of similarity. The method has also been adopted to measure a distance between marked point processes to analyze foreign currencies (Suzuki et al., 2010) and irregularly sampled palaeoclimate data (Eroglu et al., 2016). Although the existing definition of *edit distance* is quite suitable to measure similarity between event series, it introduces a bias (discussed in Sec. 3), when there are large gaps in the data as in extreme event time





series. Additionally, the method depends on multiple parameters and often it is difficult to associate a physical meaning to

them. This further complicates the parameterization of the method in case of extreme events

In this study, we propose a modification of the edit distance metric for analyzing extreme event-like time series. The proposed extension allows to consider the shifting parameter of the edit distance metric in terms of a temporal delay which can be physically interpreted as a tolerance introduced to deal with the quasi-periodic nature of a real world extreme event time series. We demonstrate the efficacy of the proposed modified edit distance measure by employing recurrence plots and their

quantification for characterizing the dynamics of flood time series from the Mississippi river in the United States. Flood time series shows a complex time-varying behavior. Moreover, flood generation is often characterized by nonlinear catchment response to precipitation input or antecedent catchment state (Schröter et al., 2015), requiring methods able to deal with nonstationarity and nonlinearity. By using the random shuffle surrogate method, we show that there is a significant serial dependency in the flood events.

**2 Edit Distance for event-like time series**

Distance measurements between two data points play an important role for many time series analysis methods, for example, in recurrence plot analysis (RP) (Marwan et al., 2007), estimation of the maximum Lyapunov exponent (Rosenstein et al., 1993), scale-dependent correlations (Rodó and Rodríguez-Arias, 2006), data classification (Sakoe and Chiba, 1978), and correlation dimension estimation (Grassberger, 1983). In case of regularly sampled data, the Euclidean distance is often used. However,

for event-like data where big gaps between events are common, this approach is not directly applicable.

Victor and Purpura (1997) introduced a method, called *edit distance*, to calculate a distance between spike trains. Later, Suzuki et al. (2010) extended this method to include changes in the amplitude of events. Ozken et al. (2015) suggested a novel interpolation scheme for irregular time series based on the edit distance, and Ozken et al. (2018) extended this approach to perform recurrence analysis for irregularly sampled data.

In order to apply the edit distance as a distance measure for, e.g., recurrence analysis, the whole time series is divided into small, possibly overlapping, segments (windows), which should contain some data points (Fig. 1). To transform one segment to another, three elementary operations are required: (1) delete or (2) insert a data point, and (3) shifting a data point to a different

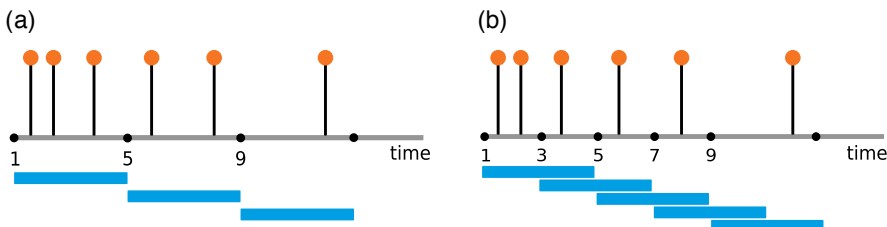

**Figure 1.** (a) Non overlapping windows of window size 4; (b) overlapping windows with 25% sharing.




point in time; each of these operations is assigned a cost. In Fig. 2, we show an illustrative example of how the elementary operations transform an arbitrary segment $S_a$ to a segment $S_b$.

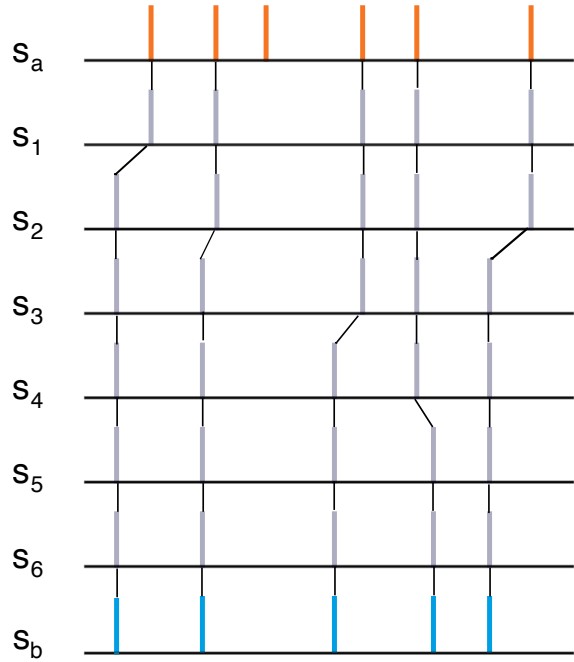

**Figure 2.** A minimum-cost path is shown to transform $S_a$ to $S_b$.

For the edit distance we use the sum of the costs of the three operations. The cost function is

$$P(C) = \Lambda_s(\,|\,I\,|+|\,J\,|-2\,|\,C\,|\,) + \sum_{(\alpha,\beta)\in C} \Lambda_0 \|t_a(\alpha) - t_b(\beta)\|,$$

where $\alpha$ and $\beta$ are events in segments $S_a$ and $S_b$ occurring at times $t_a(\alpha)$; $C$ is the set containing all such pairs $(\alpha,\beta)$ from the two segments; $I$ and $J$ are the sets of indices of events in $S_a$ and $S_b$; $|\,I\,|$, $|\,J\,|$ and $|\,C\,|$ are the cardinalities of $I$, $J$ and $C$; $\Lambda_s$ is the cost of deletion/insertion and $\Lambda_0$ the cost assigned for shifting events in time. The first summand in the cost function

deals with deletion/insertion and the second summand (the summation) deals with shifting of the pairs $(\alpha,\beta)\in C$.

The distance $D$ is the minimum cost needed to transform the event sequence in $S_a$ to the event sequence in $S_b$

$$
\begin{aligned}
D(S_a, S_b) &= \min_C P(C) \\
&= \min_C \left\{ \Lambda_s(\,|\,I\,|+|\,J\,|-2\,|\,C\,|\,) + \sum_{(\alpha,\beta)\in C} \Lambda_0 \|t_a(\alpha) - t_b(\beta)\| \right\}.
\end{aligned}
\tag{1}
$$

The operation costs $\Lambda_s$ and $\Lambda_0$ are defined as

$$\Lambda_s = \text{const., and} \tag{2a}$$






$$\Lambda_0 = \frac{M}{\text{total time}}, \tag{2b}$$

where $M$ is the number of all events in the full time series. Originally, $\Lambda_s$ was chosen as a constant value 1 (Victor and Purpura, 1997). Later, $\Lambda_s$ was optimized in a range $[0,4]$ (Ozken et al., 2018). The units of the parameters are $(\text{time})^{-1}$ for $\Lambda_0$ and unit-less for $\Lambda_s$.

The treatment of an event series as point process makes the edit distance measure a good starting point for defining a distance between segments of an extreme event time series. However, the existing form of the edit distance has a linear dependency on the difference between the occurrences of events which is inappropriate for an extreme event time series, as the rare occurrence leads to large gaps between events. Also, as already mentioned, the existing method depends on a number of parameters. Therefore, we suggest a modification of the cost function to address these two concerns.

## 100  3  Modified Edit Distance

The cost of transformation between related events (e.g. events belonging to the same climatological phenomenon in a climate event series) should be lower than that between independent, unrelated events. Hence, the shifting operation should be a more likely a choice for comparison between segments if the events in each segment are related or belong to the same phenomenon. Consequently, deletion/insertion as a choice of operation for transformation tends to be associated with unrelated events.

Now, we consider two event sequences $S_a$ and $S_b$, where each of them has only one event at times $t_a$ and $t_b$, respectively (Fig. 3). We want to transform segment $S_a$ to $S_b$ by using either deletion/insertion or shifting. The path for shifting the solitary spike has the cost $\Lambda_0 \Delta t$ where $\Delta t = \mid t_a - t_b \mid$, i.e., it grows linearly with the distance between the two events. On the other hand, the cost to delete the event at $t_a$ from $S_a$ and insert it at time $t_b$ in $S_a$ for it to resemble $S_b$ has cost 2, as the single operation cost for deletion and insertion is each 1. So the shifting operation will take place only as long as the time difference between both events is smaller then $\Delta t < \frac{2}{\Lambda_0}$.

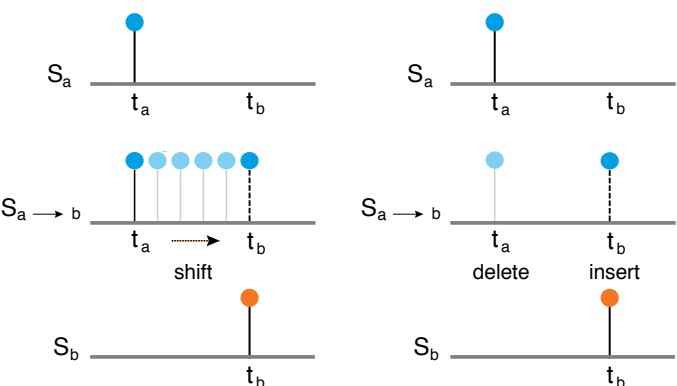

**Figure 3.** Figures illustrating the operation of shifting (left) and deletion/insertion (right).




The above condition has two limiting cases which need to be considered for an unbiased choice between shifting and deletion/insertion – when the cost of shifting is too low or when it is too high. The former case arises for an extreme event time series where the number of extreme events $M \ll$ total length of the time series, i.e., $\Lambda_0 \rightarrow 0$, Eq. (2b). This assigns a low cost for the transformation due to a biased choice of shifting over deletion/insertion for largely separated events which may be

independent.

The other limiting case, when the cost of shifting per unit time, $\Lambda_0$, is moderately high, leads to a biased preference for the deletion/insertion operation over shifting depending on how high $\Lambda_0$ is, because the cost of shifting increases linearly with the distance between two events according to the definition in Eq. (1). In this case, related events separated by a relatively small gap may be considered as independent as the shifting cost may exceed the deletion/insertion cost because of higher $\Lambda_0$. The

following example of a real extreme event series illustrates how this biased preference can lead to erroneous results. In regions with seasonal precipitation regimes, high precipitation events are more similar to those of the same season of another year compared to other seasons. However, the exact time of occurrence of the extreme precipitation events varies from year to year, i.e., even if the events in each segment are related, there might be a certain time delay between the events (Fig. 4). Now, a linear dependency of the shifting cost on the difference between times of events, Eq. (1), does not allow us to consider small delays

between potentially similar events. In addition, a not so high value of $\Lambda_0$ could increase the cost of shifting for small delays higher than the cost of deletion/insertion, implying a lower similarity between the segments.

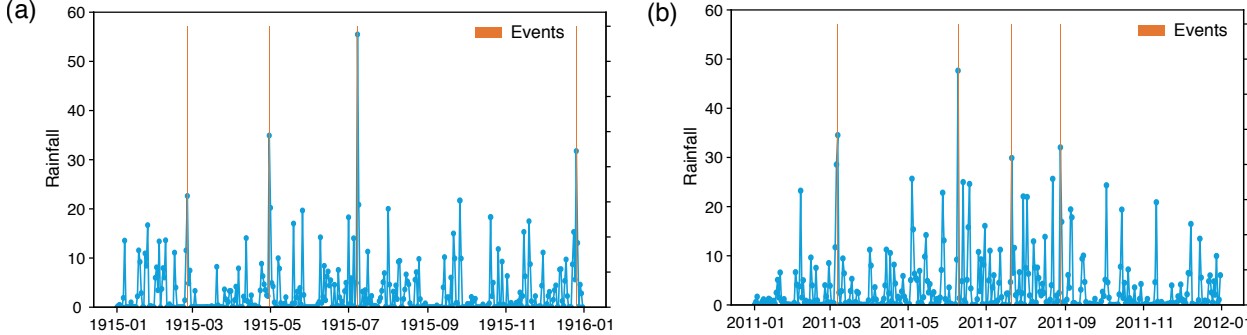

**Figure 4.** Daily precipitation time series of the Aroostook River catchment near Masardis, Maine in the United States for the years (a) 1915 and (b) 2011; red bars indicate extreme rainfall events (4 events per year), usually occurring in spring and summer, whereas in autumn extreme rainfall events usually do not occur.

As the maximum cost for shifting is limited by the cost for deletion/insertion, we need to lower the cost of shifting to get a higher similarity between the segments for the above case. This modification is done by considering the temporal distance $\Delta t$ at which the maximum cost of shifting occurs as a delay between the two events.

Selecting a certain temporal delay is relevant for climate time series, such as precipitation, or hydrological variables, like discharge, which are of an event-like nature, where the synchronization between extreme events (Malik et al., 2010, 2012;


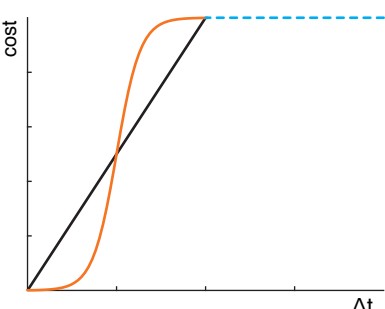

**Figure 5.** Linear cost function for shifting (black) and alternative, nonlinear cost function (orange).

Boers et al., 2013, 2014; Ozturk et al., 2018) at different geographical locations and recurrence of extreme events for the same location are of interest.

The above discussion illustrates the importance of an appropriate choice of the cost of shifting per unit time, $\Lambda_0$, based on

properties of the event series, such as time series length, total number of events, and event rate. Thus it is more reasonable to optimize $\Lambda_0$ as opposed to $\Lambda_s$. Victor and Purpura (1997) suggested generalizations of the cost assigned to finite translations, such as modifying the simple $\Lambda_0 \Delta t$ to more general functions, satisfying the triangle inequality.

In view of comparing two events under a predefined delay, we suggest to replace the linear cost function for shifting, Eq. (1), by a nonlinear cost function which allows a temporal tolerance and also ensures a smooth change from the cost function for

shifting to the cost for deletion/insertion (Fig. 5). We propose to use the Logistic function (Cramer, 2002) as the cost for shifting:

$$f(t) = \frac{\Lambda_s}{1 + e^{-k(t-\tau)}}, \tag{3}$$

where

- $\tau$ is the chosen time interval between events marking the transition of the function from exponential growth (for closely-

spaced events) to bounded exponential growth (for events separated by large time gaps) (Fig. 6a),

- $\Lambda_s$ is the maximum cost, and

- $k$ is a parameter that affects the rate of the exponential growth, in this study we choose $k = 1$. For $k = \infty$, the Logistic function becomes a step function.

The Logistic function (Eq.(3)) is used for modeling the population growth in an area with limited carrying capacity. The

sigmoid nature of the Logistic function is apt for representing the cost of shifting with low values for closely-spaced events and which tends to saturate for events separated by a time delay greater than a certain $\tau$.




A step function (e.g., Heaviside function) would be another choice for the cost function according to whether shifting is chosen below a certain delay and deletion/insertion is chosen above it (Fig. 6b):

$$f(t) = \begin{cases} \Lambda_0 & \Delta t \leq \tau \\ \Lambda_s & \Delta t > \tau \end{cases} \tag{4}$$

where $\Lambda_0$ is the cost for shifting, according to Eq. (2b), and $\tau$ is the given delay choice which decides between shifting and deletion/insertion. However, such a cost function maintains a constant value irrespective of the event-spacing and switches to a different value only at the delay threshold. Therefore, the logistic function, as a smooth approximation of the step function, is a preferable choice.

The parameter optimization problem of the modified edit distance equation now becomes a problem of solving a single
linear equation with two unknown variables – the coefficient related to the cost of shifting events in time and $\Lambda_s$. Since, in our method, we choose to optimize the former by using the Logistic function, we keep $\Lambda_s$ constant. This constant can be absorbed in the optimization function of the first term, and therefore, the coefficient of the second term related to the cost of addition/ deletion.

For a particular pair of events $(\alpha, \beta)$, we can arbitrarily set the maximum cost $\Lambda_s$ for shifting or deletion/insertion to be 1,
because it is the only free cost parameter (we have no $\Lambda_k$ because of neglecting the amplitudes). Thus, the cost for transforming one segment to another using the *modified edit distance* (mED) is defined as

$$P(C) = (\mid I \mid + \mid J \mid - 2 \mid C \mid) + \sum_{(\alpha,\beta) \in C} \left\{ \frac{1}{1 + e^{-k(\|t_a(\alpha) - t_b(\beta)\| - \tau)}} \right\}$$

and the corresponding distance function to calculate the distance between two segments $S_a$ and $S_b$ is

$$D(S_a, S_b) = \min_C \left\{ (\mid I \mid + \mid J \mid - 2 \mid C \mid) + \sum_{(\alpha,\beta) \in C} \left\{ \frac{1}{1 + e^{-k(\|t_a(\alpha) - t_b(\beta)\| - \tau)}} \right\} \right\}. \tag{5}$$

This new definition of cost depends only on the parameter $\tau$, which can be interpreted in the sense of a delay between events. Moreover, this definition holds the triangular inequality when $\tau$ satisfies a certain condition (see the Appendix for proof).

In the next section we use the modified version of *edit distance* to perform recurrence analysis of extreme event-like data.

## 4   Recurrence plots

A recurrence plot (RP) is a visualization of the recurrences of states of a dynamical system, capturing the essential features
of the underlying dynamics into a single image. The quantification of the patterns in a RP by various measures of complexity provides further (quantitative) insights into the system's dynamics (recurrence quantification analysis, RQA) (Marwan et al., 2007). RQA was designed to complement the nonlinearity measures such as the Lyapunov exponent, Kolmogorov entropy, information dimension, and correlation dimension (Kantz, 1994). RP-based techniques have been used in many real world problems from various disciplines. In the field of finance, Strozzi et al. (2002) studied RQA measures for high frequency
currency exchange data. In astrophysics, Stangalini et al. (2017) applied RQA measures to detect dynamical transitions in solar


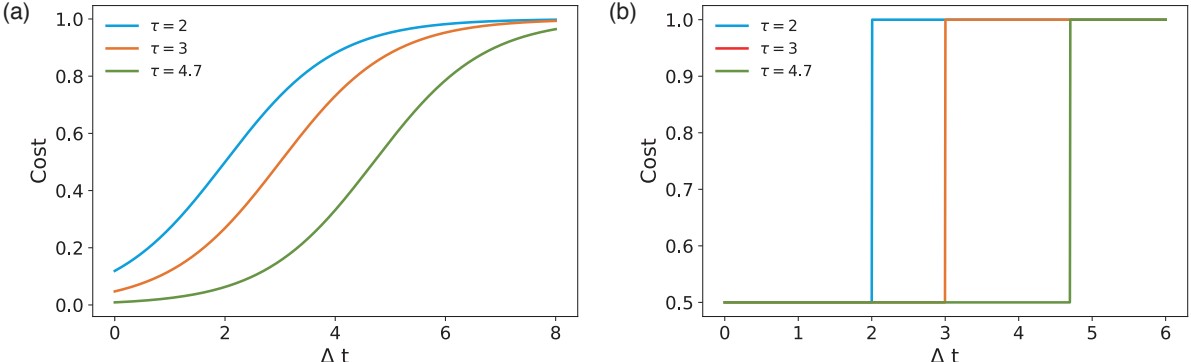

**Figure 6.** (a) Logistic function with $k = 1$ and (b) Heaviside function for delay $\tau$ equal to $2$, $3$ and $4.7$, respectively. $\Delta t$ is the gap between two events in kept up to the maximum value $6$.

activity in the last 150 years. It has also been used to classify dynamical systems (Corso et al., 2018), or to detect regime changes (Marwan et al., 2009; Eroglu et al., 2016; Trauth et al., 2019) and has many applications in Earth science (Marwan et al., 2003; Chelidze and Matcharashvili, 2007), econophysics (Kyrtsou and Vorlow, 2005; Crowley and Schultz, 2010), physiology (Webber, Jr. and Zbilut, 1994; Zbilut et al., 2002; Marwan et al., 2002; Schinkel et al., 2009), and engineering (Gao
et al., 2013; Oberst and Tuttle, 2018).

Consider a time series that encodes $N$ measured states of a dynamical system $\{\boldsymbol{x}_i \in M\}_{i=1}^{N}$ on space $M$. A state of this system is said to be recurrent if it falls into a certain $\varepsilon$-neighborhood of another state. Given a distance function $D : M \times M \rightarrow \{0\} \cup \mathbf{R}^{+}$, for a given trajectory $\boldsymbol{x}_i$, the *recurrence matrix* of the system is defined as:

$$R_{i,j}(\varepsilon) = \begin{cases} 1, & \text{if } D(\boldsymbol{x}_i, \boldsymbol{x}_j) < \varepsilon, \\ 0, & \text{otherwise.} \end{cases} \tag{6}$$

In a RP, when $R_{i,j}(\varepsilon) = 1$ a point is plotted at $(i, j)$, otherwise nothing is plotted. Different classes of dynamics result in different patterns in their respective RPs (Marwan et al., 2007).

The RP contains a main diagonal line, called the line of identity (LOI), corresponding to the recurrence of a state with itself. The RP is symmetrical about the LOI when $D$ is a metric (e.g., a symmetrical norm). We are interested in the line structures in RP as they capture several aspects of the underlying dynamical behaviour of a system. For instance, long, continuous lines parallel to the LOI denote (pseudo-)periodic behavior, whereas short, discontinuous diagonal lines are indicative of a chaotic
system.

In order to incorporate mED in RP, we divide the time series into small segments and compute the distance between these segments; the time indices are the centre points of these segments. We can now define the RP in terms of the distance between the segments calculated by mED:

$$R_{i,j}(\varepsilon) = \Theta(\varepsilon - D(S_i, S_j)) \quad i, j = 1, 2, ..., \omega, \tag{7}$$

where $\omega$ is the number of segments and the size of the RP is $\omega \times \omega$.





One of the most important measures of RQA is the determinism (DET), based on the diagonal line structures in the RP. The diagonal lines indicate those time periods where two branches of the phase space trajectory evolve parallel to each other in the phase space. The frequency distribution $P(l)$ of the lengths of the diagonal lines is directly connected to the dynamics of the system (Marwan et al., 2007). The ratio of the recurrence points which form diagonal lines to all points on the RP is the measure of *determinism*

$$DET = \frac{\sum_{l=2}^{N} l P(l)}{\sum_{i,j}^{N} R_{i,j}}. \tag{8}$$

RPs of stochastic processes mainly contain single points, resulting in low DET values, where RPs of deterministic processes contain long diagonal lines, resulting in high DET values.

In this work, we are interested in the deterministic nature of flood event time series. For this purpose, we focus on the DET measure.

## 5 Choice of window size

We divide the time series into segments/windows. If adjacent windows overlap, we call it overlapping sliding window, else, non-overlapping window (Fig. 1). Sliding window techniques are widely used for signal processing (Bastiaans, 1985), activity recognition processes (Dehghani et al., 2019). The window size should be selected properly, i.e. each window should contain enough data points to be differentiable from similar movements.

Consider a time series of data $x_i \in \mathbb{R}$ at times $t_i \in \mathbb{N}$. For generalization we consider a constant sampling rate, i.e.,

$$\Delta T = t_{i+1} - t_i = \text{const.} \ \forall i \in \mathbb{N}.$$

Each window consists of $n$ ($n \in \mathbb{N}, n > 1$) data points, so the window size is $w = n\Delta T$. For overlapping windows, a fraction of the data is shared between consecutive windows, denoted by $L \in \{1, 2, 3, .., n-1\}$ as the number of data points within the overlapping range, where $L = \varnothing$ signifies the non-overlapping case. The overlapping range is $OL = L\Delta T$ and in terms of percentage $OL(\%) = \frac{L}{n}$. The number of data points being shared for overlapping windows is $p = w\Delta T - OL$.

Because mED focuses on event-like data, and finding an optimum window size is important. An inappropriate window size may lead to missing events in windows ("empty windows") due to the sparse occurrence of events. Here we propose some criteria for choosing the optimum window size:

– To avoid empty windows we try to fix the number of events $(n)$ in each window. Here we choose the window size as 1 year, as most climate phenomena such as flood events exhibit annual periodic behaviour and therefore we get similar number of events each year.

– In case we need to choose a larger window size to have enough data points in each window, the number of segments decreases if the windows do not overlap which in turn reduces the dimension of the recurrence matrix. The underlying





dynamical behaviour of the system will not be completely captured by the resulting coarse-grained recurrence plot. Overlapping windows alleviate this problem by increasing the number of windows. In our work, we use a certain percentage of overlapping.

– An inappropriate window size can lead to the aliasing effect (Fig. 11). As a result different signal would be indistinguishable and we might loose important transitions.

## 6 Applications of modified edit distance

We apply mED first to generated data and then to real-world flood observations to understand how well our method can identify recurrences in extreme event-like time series.

### 6.1 Numerically generated event-series using Poisson process

The Poisson process is used as a natural model in numerous disciplines such as astronomy (Babu and Feigelson, 1996), biology (Othmer et al., 1988), ecology (Thomson, 1955), geology (Connor and Hill, 1995), or trends in flood occurrence (Swierczynski et al., 2013). It is not only used to model many real-world phenomena but also allows for a tractable mathematical analysis.

Consider $N(t)$ to be a stochastic counting process which represents the number of events above some specified base level in the time period $(0, t)$. Suppose the events occur above the base level at a constant rate $\lambda > 0$ (units of 1/time). So, the probability that $n$ events occur in the time between $t$ and $t + s$ is given by (Ross, 1997; Loaiciga and Mariño, 1991)

$$P\{N(t+s) - N(s) = n\} = e^{-\lambda t} \frac{(\lambda t)^n}{n!} \quad (n = 0, 1, 2, \ldots). \tag{9}$$

A Poisson process is a set of random events whose stochastic properties do not change with time (stationary) and every event is independent of other events, i.e., the waiting time between events is memory-less (Kampen, 2007).

Here, we study three cases of numerically generated event series which are motivated by the occurrence of natural events. First, we test mED for a simple Poisson process. Each subsequent case tries to capture features of these real world phenomena by adding an element of and memory to the simple Poisson process. We compare the RPs and the structures in the RPs (by considering the RQA measure DET) derived from the standard edit distance and from the modified edit distance.

For ED (Eq. 1), the cost parameter for shifting is calculated according to Eq. (2a) and $\Lambda_s = 1$, whereas Eq. (4) is used for mED, recurrence threshold $\varepsilon = 10$ percentile of the distance matrix for each case. We choose the upper bound for the range of $\tau$ to be less than the mean inter-event time gap for the complete time series. The physical interpretation of this choice is that the temporal tolerance or delay in the arrival of events in a particular season (event cluster) should not only be less than the time period of the seasonal cycle but also relatively less than the length of the season.

It is expected that the quasi-periodic behaviour of the event series will lead to high DET values. In case of ED, the DET should be constantly high at all $\tau$, as ED does not include the concept of time delay. On the other hand, DET computed using mED should first increase with $\tau$ and then slowly decrease.

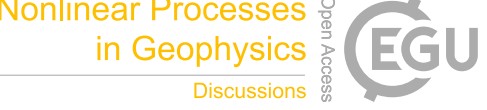



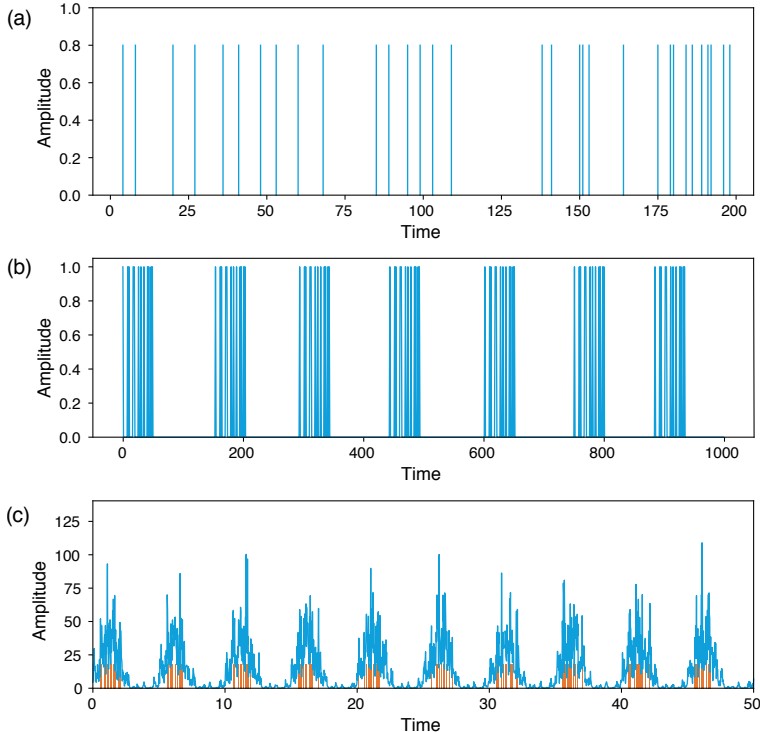

**Figure 7.** Generated time series from a (a) homogeneous Poisson process (total length =10,000), (b) repeating Poisson process (total length=22,442), and (c) Poisson process with periodical forcing to mimic discharge time series (total length 0–500 with 10,000 equally spaced points), small red bars indicate the events.

.

### 6.1.1 Homogeneous Poisson process

The homogeneous Poisson process models the occurrence of random spaced, i.e. stochastic, events in time, where the average time between the events is known. It is used to model shot noise, radioactive decay, arrival of customers at a store, earthquakes, etc. Here we construct an event series (Fig. 7a) from Eq. (9) with $\lambda = \frac{1}{5}$. The RP of a realisation of such a homogeneous Poisson process is shown in Fig. 8a,b. In a stochastic process, a recurrence of a randomly selected state occurs by chance, resulting in randomly distributed points in the RP. Accordingly, DET has low values (Fig. 8c).

### 6.1.2 Repeating Poisson process

We generate a small segment of a simple Poisson process and repeat the same segment after certain time gaps (Fig. 7b). Here, the Poisson process is generated for a segment of length 50 with mean rate of events $\lambda = \frac{1}{5}$. The gap between the event segments is chosen randomly in the range 105 to 115. The RP of this event series is shown in Fig. 9. Identical sequences of events result in longer diagonal lines in the RP (with lengths in the order of the event sequences), but the varying time gaps between the event




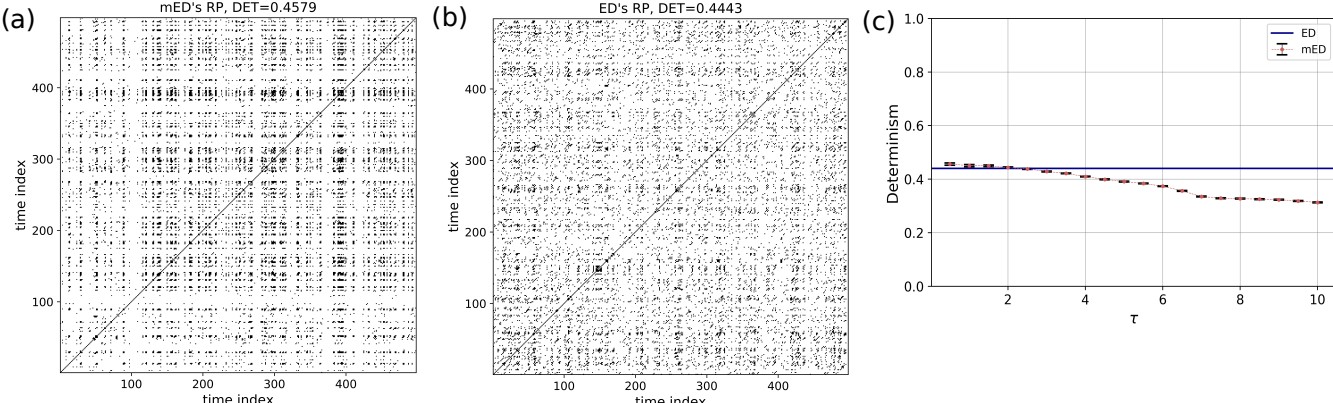

**Figure 8.** RP of a homogeneous Poisson process (Fig. 7a) using (a) mED, (b) ED; (c) comparison of DET for 300 realisations using ED (blue horizontal line) and using mED for varying $\tau$ in range 0 to 10.

chunks make the diagonal lines discontinuous implying a quasi-periodic behaviour. Quasi-periodicity is often a characteristic of an extreme event time series such as flood events. RP using ED Fig.9b contains more short and discontinuous diagonal lines

corresponds to less DET value. In Fig.9c, we find a certain range where the DET value calculated using mED is higher than ED.

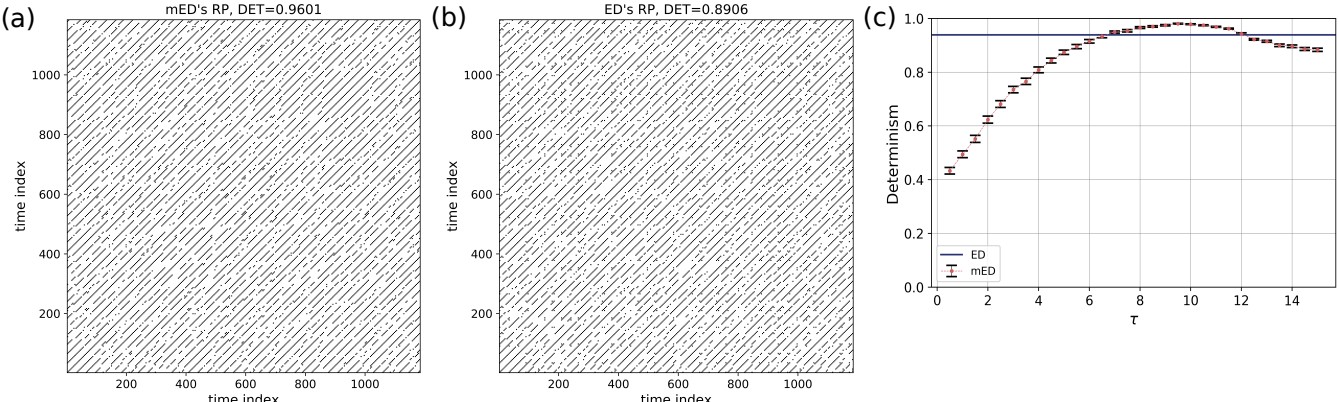

**Figure 9.** RP of a repeating Poisson process (Fig.7b) using (a) mED, (b) ED; (c) comparison of DET for 300 realisations using ED (blue horizontal line) and using mED for varying $\tau$ in range 0 to 15.

### 6.1.3   Poisson process with periodical forcing

Quasi-periodicity in a time series can be expressed as a sum of harmonics with linearly independent periods. Here we construct a time series with superposition of a slow moving signal $f(T_1) = \sin(2\pi t/T_1)$ and a fast moving signal $f(T_2) = \sin(2\pi t/T_2)$,

whereas the events are drawn by a Poisson process (Fig. 7c). We select the extreme events by using a certain threshold and





take the peak over the threshold (POT). Their occurrences and positions can be seen as the outcome of a point process (Coles, 2001). Such a time series can be used to mimic a streamflow time series which may exhibit periodic behaviour on several scales (annual, seasonal, decadal, etc.). Here the window size is equal to the time period of the slow moving signal. The RP is shown in Fig. 10. The periodic occurrences of the event chunks is visible in the RP by the line-wise accumulation of recurrence

points with a constant vertical distance (which corresponds to the period). The stochastic nature of the "local" event pattern is visible by the short diagonal lines of varying lengths. As mentioned earlier, when using an improper window size we can lose recurrence points and a smaller number of diagonal lines occurs due to the aliasing effect (Fig. 11).

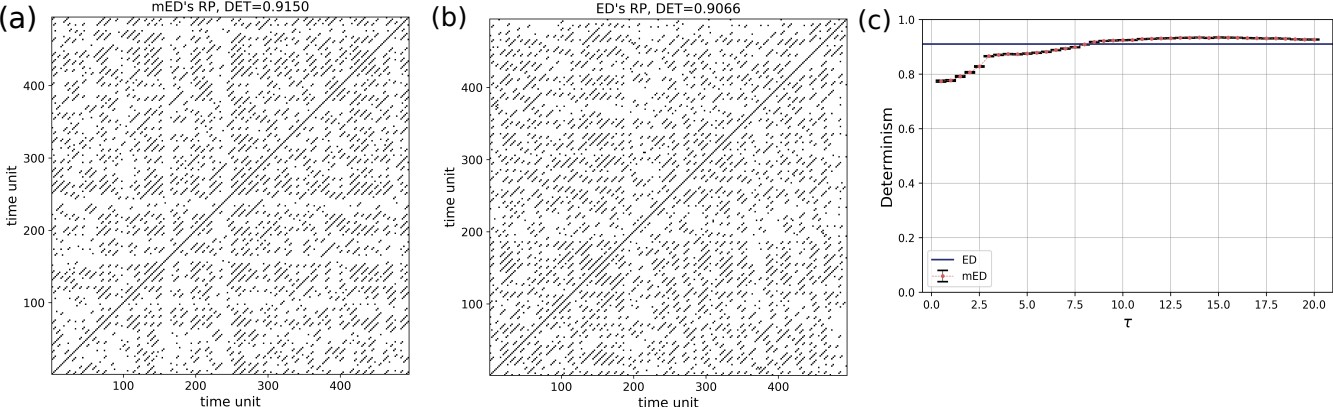

**Figure 10.** RP of Poisson process with periodical forcing (Fig.7c) using (a) mED, (b) ED; (c) comparison of DET for 300 realisations using ED (blue horizontal line) and using mED for varying $\tau$ in range 0 to 20.

For a certain range of $\tau$, DET for mED (Fig.8, Fig.9, and Fig.10) to be higher than the DET for ED, thus capturing the underlying periodic behaviour better.

**6.2  Recurrence analysis of flood events**

As a case study, we apply our method to the Mississippi river which has a rich history of flood events. Here we are motivated to study the recurring nature of flood events using recurrence plots. Recurrence plot analysis also helps to quantify the serial dependency of flood time series. (Wendi et al., 2019) used RQA to study the similarity of flood events.

We use the mean daily discharge data of the Mississippi River from the Clinton, Iowa station in the USA for the period

1874–2018. The data is obtained from the Global Runoff Data Centre (www.bafg.de/GRDC).

We construct the flood event series from the complete discharge time series as follows: We first apply a threshold (corresponding to the 99[th] percentile) for each year on the discharge values which gives a few events (say 3/4) per year. If several successive days fall above the threshold, we pick only the maximum event from this cluster. Then we lower the threshold by 0.1 percentiles (the threshold typically lies between 99-90 percentile) and repeat the above step until we get the desired number

of independent events.

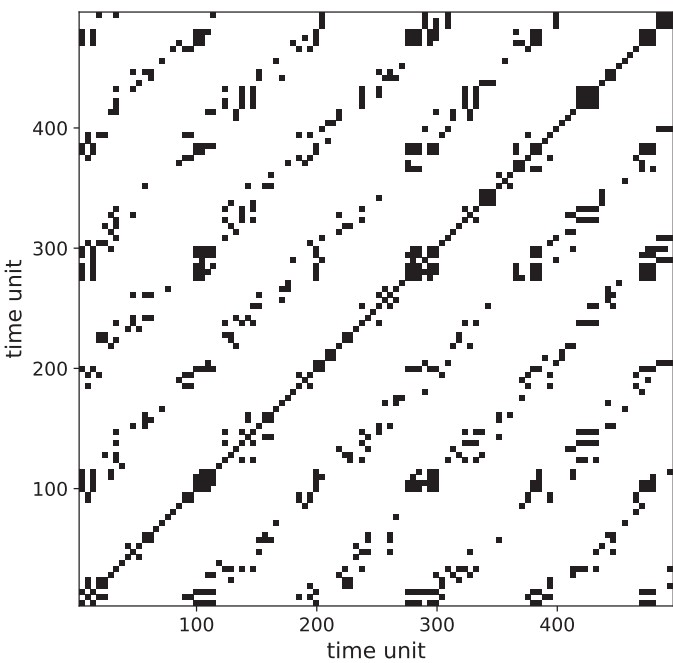

**Figure 11.** RP of Poisson process with periodical forcing (Fig. 7c) using mED showing aliasing effect due to improper window size (note that this effect also occurs for the standard ED approach).

We apply mED for finding recurrence patterns for flood events. To this end, we choose our window size equal to the annual cycle (1 year) with 6 months and 9 months of overlapping and with the cost function of delay, $\tau = 15, 30, 45$ and days. The recurrence threshold $\varepsilon = 8$ percentile of the distance matrix. The recurrence plot of flood events is shown in Fig. 12 and 13 . We also compute the recurrence plot using ED and calculate DET Fig. 14.

The black points in the RP (Fig. 12, 13) denote the segments which have higher similarity and the white points imply less similarity in the occurrence of flood events. The RP of Mississippi floods shown in Fig 13d might look similar to the RP in Fig. 8 for events generated by Poisson process. However, the zoomed in image of the RP of Mississippi floods, Fig 13d, is very similar to the RP in Fig. 10a for events generated by Poisson process with periodical forcing. Diagonal structures are seen in the flood RP ( Fig. 12, 13) denoting an inherent serial dependency in the data as indicated by the DET value. Serial dependence

is a property by virtue of which the future depends on the past. When diagonal lines are likely to appear in a RP, the current neighbors tend to be neighbors in the near future, and thus, the corresponding system is said to have serial dependence.

To determine the statistical significance of the RP analysis, we develop the following statistical test. We use random shuffle surrogates (Scheinkman and LeBaron, 1989) statistical testing. Suzuki et al. (2010) used this method for finding serial dependency on foreign exchange tick data by quantifying the diagonal lines in the RP (DET measure).

We first set a null hypothesis, then we generate a set of random surrogates that preserve the null hypothesis property. After that, we compare the test statistics of the original data with the surrogates data. We can reject the null hypothesis if the test


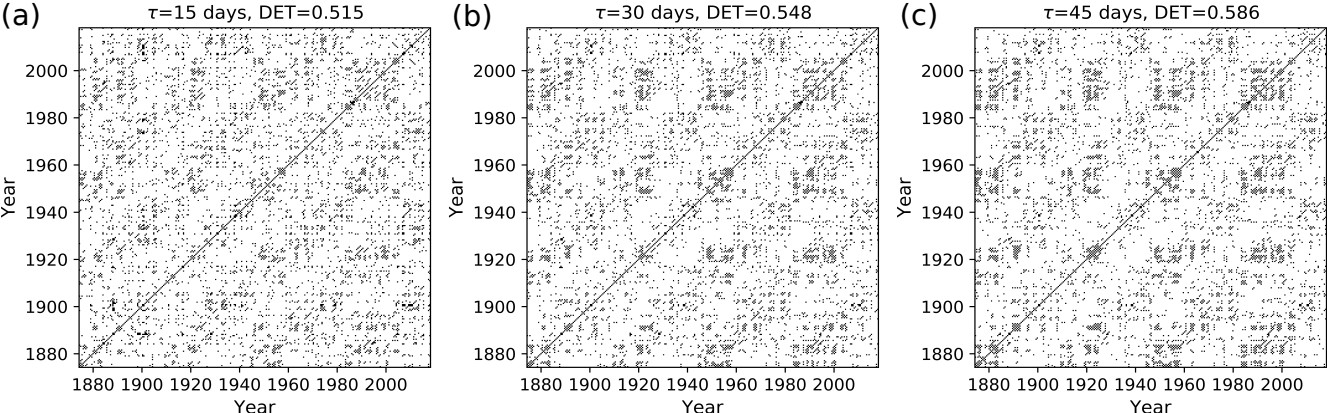

**Figure 12.** Recurrence plot of flood events using mED, window size = 1 year with 6 months overlap for delay $\tau$ = (a) 15 days, (b) 30 days, and (c) 45 days.

statistics obtained from the surrogate data and the original data is out of the specified range, else we cannot reject the null hypothesis.

The null hypothesis is that there is no serial dependency in the data. Thus, we expect that after each random shuffling the
information will be preserved. We create each random surrogate by randomly shuffling rows and columns of a recurrence plot simultaneously. We can reject the null hypothesis if the test statistics of the original data $K_0$ (DET value) is out of the specified range from the distribution of those surrogate datasets $K_S$ (Theiler et al., 1992). The algorithm works as follows:

1. Calculate the distance matrix and compute the RP for a certain window size.

2. Reorder the row and columns randomly to create surrogate recurrence plots.

3. Calculate the DET of the surrogate recurrence plots.

4. Repeat the steps 1–3 and get DET for 50 different surrogate RPs.

We assume the statistics obtained from the surrogates $K_s$ is normally distributed and we consider only one experimental data set. The measure of "significance" as defined by Theiler et al. (1992):

$$\rho = \frac{K_0 - \mu_{K_s}}{\sigma_{K_s}} \tag{10}$$

follows a $t$-distribution, the number of surrogates are $n = 50$ with $(n-1)$ degrees of freedom. $\mu_{K_s}$ and $\sigma_{K_s}$ are the mean and standard deviation of the statistics of surrogate data. The value of $\rho$ has to be compared to the value of the $t$-distribution that corresponds to the 99[th] percentile and the mentioned degrees of freedom, i.e., $t_{0.01/2}(49) = 2.68$. If $\rho$ exceeds this value, the null-hypothesis has to be rejected.







**Figure 13.** Recurrence plot of flood events using mED, window size = 1 year with 6 months overlap for delay $\tau$ = (a) 15 days, (b) 30 days, and (c) 45 days; (d) zoom in image of the blue area of (c).

We measure $\rho$ for the data generated from a homogeneous Poisson process, Eq. (9), and also for the flood event data. First, for the Poisson process the value of $\rho = 1.91, < 2.68$, so we cannot reject the null hypothesis, confirming the missing serial dependency in the homogeneous Poisson process.


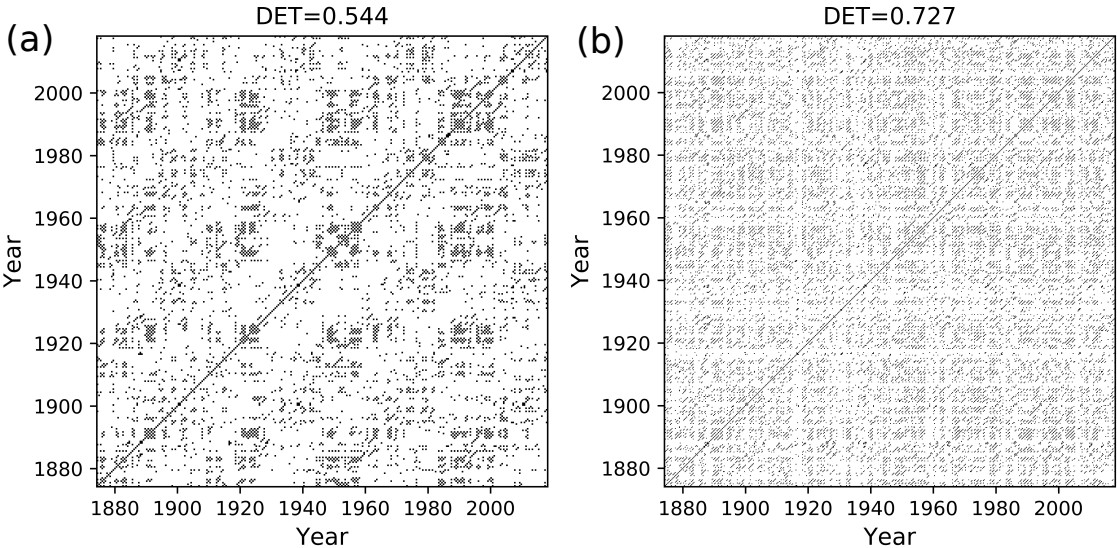

**Figure 14.** Recurrence plot of flood events using ED, window size = 1 year, with (a) 6 months overlapping and (b) 9 months overlapping.

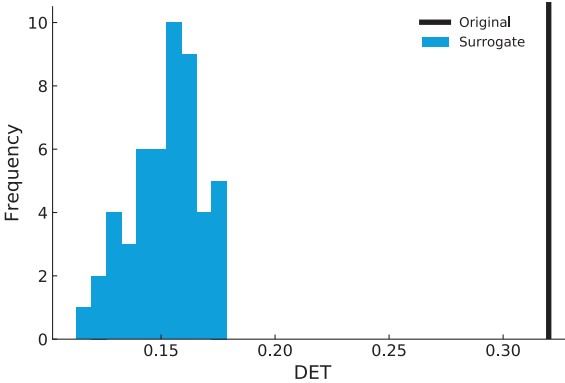

**Figure 15.** Distribution of DET value of surrogate data (blue) and the original DET value (black).

For the flood event data, the distribution of the test statistics (DET) is far away from the value of the original data (Fig. 15). The $\rho$ value for the flood event RP with 6 months overlapping and $\tau = 15$ days is 23.80 which is much larger than 2.68. Hence, we can reject the null hypothesis at the significance level of $\alpha_0 = 0.01$, getting strong indications that there is serial dependency at high significance level in the occurrence of flood events.





## 7  Conclusion

In this paper, we propose a distance measure for recurrence based analysis of extreme event time series. The proposed measure is based on a modification of the *edit distance* measure proposed by Victor and Purpura (1997). We include the concept of time delay to incorporate the slight variation in the occurrence of recurring events of a real world time series due to changes in

seasonal patterns by replacing the linear dependency of the cost of shifting events by a nonlinear dependency using the logistic function. This is substantial improvement over the previous definition of the edit distance as used for the TACTS algorithm proposed by Ozken et al. (2015) as the optimization of $\Lambda_0$ is based on the temporal delay between events, which is of physical relevance to the study of extreme events and can be chosen according to the phenomenon being studied. The modified edit distance also reduces the number of independent parameters. We tested mED on prototypical event series generated by a point

process (Poisson process). We found that for the quasi-periodic repeating Poisson process event series, the determinism of RP computed using mED varies with temporal delay and is higher than the measured DET value of RP computed using ED for a certain range of delay. We applied the method to study recurrences in the flood events of the Mississippi river. Our analysis revealed deterministic patterns in the occurrence of flood events from RP. Finally, using the random shuffle surrogate method we have shown that the data of the occurrence of flood events has a statistically significant serial dependency.

In this work, we have only considered binary extreme event time series, and ignored the amplitude of events. Next, the mED measure should be further modified by including the cost due to difference in amplitude for applications in real world time series where patterns of intensity of extremes might be of interest. Moreover, an elaborate method to optimize for a proper window size may be devised to capture recurrences of events at different time scales.

## Appendix A

For deletions and insertions, the new definition inherits the triangle inequality for the original definition of edit distance by Victor and Purpura (1997). Thus, we can show that the new definition of the edit distance preserves the triangle inequality as a total if a shift preserves the triangle inequality. For this sake, considering the following situation is sufficient: Suppose that there are three simple point process each of which has an event; the first point process has an event at $u(u > 0)$; the second point process has an event at $u + s(s > 0)$; the third point process has an event at $u + s + t(t > 0)$. Therefore, the first and the

second point processes have an inter-event interval of $s$; the second and the third point processes have an inter-event interval of $t$; and the first and the third point processes have an inter-event interval of $s + t$.

We assume that $\tau \leq \frac{s+t}{2}$ . Then, the distance between the first and the second point processes is

$$D(S_1, S_2) = \frac{1}{1 + e^{-s+\tau}}.$$

Likewise, the distance between the second and the third point processes is

$$D(S_2, S_3) = \frac{1}{1 + e^{-t+\tau}}.$$





The distance between the first and the third point processes is

$$D(S_1, S_3) = \frac{1}{1 + e^{-s-t+\tau}}.$$

Then the following chain of inequalities holds:

$$
\begin{aligned}
& -D(S_1, S_3) + D(S_1, S_2) + D(S_2, S_3) \\
=\ & -\frac{1}{1+e^{-(s+t)+\tau}} + \frac{1}{1+e^{-s+\tau}} + \frac{1}{1+e^{-t+\tau}} \\
=\ & \frac{1}{\{1+e^{-(s+t)+\tau}\}\{1+e^{-s+\tau}\}\{1+e^{-t+\tau}\}} \\
& \times \{ -(1+e^{-s+\tau})(1+e^{-t+\tau}) + (1+e^{-(s+t)+\tau})(1+e^{-t+\tau}) + (1+e^{-(s+t)+\tau})(1+e^{-s+\tau}) \} \\
=\ & \frac{1}{\{1+e^{-(s+t)+\tau}\}\{1+e^{-s+\tau}\}\{1+e^{-t+\tau}\}} \\
& \times \{ -1 - e^{-s+\tau} - e^{-t+\tau} - e^{-s-t+2\tau} + 1 + e^{-(s+t)+\tau} + e^{-2t-s+2\tau} + e^{-t+\tau} + 1 + e^{-(s+t)+\tau} + e^{-2s-t+2\tau} + e^{-s+\tau} \} \\
=\ & \frac{1}{\{1+e^{-(s+t)+\tau}\}\{1+e^{-s+\tau}\}\{1+e^{-t+\tau}\}} \\
& \times \{ -e^{-s-t+2\tau} + e^{-(s+t)+\tau} + e^{-2t-s+2\tau} + 1 + e^{-(s+t)+\tau} + e^{-2s-t+2\tau} \} \\
=\ & \frac{1}{\{1+e^{-(s+t)+\tau}\}\{1+e^{-s+\tau}\}\{1+e^{-t+\tau}\}} \\
& \times \{ e^{2\tau}(-e^{-s-t} + e^{-2t-s} + e^{-2s-t}) + 2e^{\tau}e^{-s-t} + 1 \} \\
=\ & \frac{1}{\{1+e^{-(s+t)+\tau}\}\{1+e^{-s+\tau}\}\{1+e^{-t+\tau}\}} \\
& \times \{ e^{2\tau}(-e^{-s-t} + e^{-2t-s} + e^{-2s-t} - e^{-2s-2t}) + \{e^{\tau}e^{-s-t} + 1\}^2 \} \\
=\ & \frac{1}{\{1+e^{-(s+t)+\tau}\}\{1+e^{-s+\tau}\}\{1+e^{-t+\tau}\}} \\
& \times \{ \{e^{\tau}e^{-s-t} + 1\}^2 + e^{2\tau}e^{-s-t}(-1 + e^{-t} + e^{-s} - e^{-s-t}) \} \\
=\ & \frac{1}{\{1+e^{-(s+t)+\tau}\}\{1+e^{-s+\tau}\}\{1+e^{-t+\tau}\}} \\
& \times \{ \{e^{\tau}e^{-s-t} + 1\}^2 - e^{2\tau}e^{-s-t}(1 - e^{-t})(1 - e^{-s}) \} \\
=\ & \frac{1}{\{1+e^{-(s+t)+\tau}\}\{1+e^{-s+\tau}\}\{1+e^{-t+\tau}\}} \\
& \times e^{2\tau-2s-2t}\{ \{1 + e^{-\tau+s+t}\}^2 - (e^t - 1)(e^s - 1) \} \\
\geq\ & \frac{1}{\{1+e^{-(s+t)+\tau}\}\{1+e^{-s+\tau}\}\{1+e^{-t+\tau}\}} \\
& \times e^{2\tau-2s-2t}\{ e^{-2\tau+2s+2t} - e^t e^s \} \\
\geq\ & \frac{1}{\{1+e^{-(s+t)+\tau}\}\{1+e^{-s+\tau}\}\{1+e^{-t+\tau}\}} \\
& \times \{ 1 - e^{2\tau-s-t} \} \\
\geq\ & 0
\end{aligned}
$$

As a result, we have

$$D(S_1, S_3) \geq D(S_1, S_2) + D(S_1, S_3).$$

Thus, the triangle inequality holds if $\tau \leq \frac{s+t}{2}$.



*Author contributions.* AB, BG, NM, and DE developed the theoretical formalism. AB carried out the experiment. YH verified the triangular inequality property. BM and JK closely supervised the work. AB took the lead in writing the manuscript. All authors provided critical feedback and helped shape the research, analysis and manuscript.

*Competing interests.* The authors declare that they have no conflict of interest.

*Acknowledgements.* This research has been funded by the Deutsche Forschungsgemeinschaft (DFG) within graduate research training group GRK 2043/1 "Natural risk in a changing world (NatRiskChange)" at the University of Potsdam, the BMBF funded project climXtreme. The research of YH is supported by JSPS KAKENHI (JP19H00815) and the research of DE is supported by TÜBITAK (grant 118C236) Kadir Has University internal Scientific Research Grant (BAF). AB would like to thank Matthias Kemter for valuable discussions.



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
