# Peer review of "Recurrence analysis of extreme event like data"

_Nonlinear Processes in Geophysics, 2020_

## Referee Comment (RC1) · Carsten Brandt (Referee) · 22 Nov 2020

General comments: The authors present a paper with a distance measure and its modification (the latter developed by the authors) to compare windows out of non-equally spaced time series of extreme events using recurrence quantification analysis (RQA). The paper is in general well written and fulfills from my point of view standards on scientific publications. I have several general and technical comments that should be solved/ reacted to before final publication.

Specific comments: - I must confess (and this is almost a general comment) that it was at least in first reading difficult for me to understand the edit distance (or its modification). Maybe it is good to make the basic concept clearer: in the paragraph starting at line 75, I propose to write after the first sentence that 1) the aim is to determine a distance measure between each segment and 2) that the transformation is a means to

an end, that is, the aim is creating a cost of the transformation as the distance measure. The aim is not the transformation itself. For me, this clarification/underlining is essential.

- Then, I could not find the term "edit distance" in the original paper from Victor and Purpura 1997. I think it would be good to make a remark for readers wanting to dive into the origins. - Line 254: did you really use eq. (4)? That is the Heaviside function, and you claim in chapter 3 that you use the logistic function. - Paragraph from line 296 on: I don't understand this completely. You apply a threshold on the data of daily discharge? You determine the percentile per year? A few events - you mean 3 to 4 (you wrote 3/4 = three quarters)? - Line 303 recurrence threshold: this parameter is quite crucial in RQA. You set it to 8% of the distance distribution. Is there any specific reason for this value (in the Poisson events you used 10%), was there a reasoning or trials to get to this value? Then please include these in the paper. If it was "just chosen", then write this as well please. Also important: I am missing a discussion of the difference of your modified edit distance and the original one on the Mississippi flood events, i.e., comparing figure 12 and 13 with figure 14. You mention that you computed standard edit distance and figure 14 in line 304, but I am missing a discussion about the outcome of this, or the comparison with your modified distance. A final remark: what do you think about modifying the title or amending it by "...using a modified distance measure"? This is from my point of view the main and important contribution you make. But I leave this up to you, the authors.

Technical corrections (minor contents or grammar): Line 60/61: I suggest writing either "A flood time series shows..." or "Flood time series show..." Line 67: To be strict, "RP" is the abbreviation of "recurrence plot"; "recurrence plot analysis" (RPA) is also sometimes used, but "recurrence quantification analysis" (RQA) is much more common. Line 82: "where alpha and beta... at times ta(alpha)" - isn't there "and tb(beta" missing? The definition of determinism DET has been done with a minimum line length of 2, as one can see in eq. (8). Please consider to write down "with a minimum line

length of 2" or similar to make this clear. Other line length are also used in the literature for RQA. Figure 6 caption "Deltat is the gap between two events in kept up to the maximum value 6." - I do not understand this sentence. Line 221: The symbol OL for the overlapping range might be confusing, especially in an equation in which on the right hand side (LdeltaT) L and deltaT are multiplied (I was wondering in the first place whether O shall be multiplied by L on the left hand side). Consider using an index O_L (sorry for not being able to display it here). Line 222: I do not understand the last sentence: the number of shared data points is L, is it not? This can be expressed of course as OL/deltaT. omega times deltaT minus OL makes no sense anyway, because the first summand has unit time squared and the second summand has unit time. Line 223: I suggest removing the "and". Line 234: "...different signal"s - "s" is missing Line 306 a dot after "Fig" is missing Line 306 and 3ß7: you wrote Fig. 13d twice. I guess the first shall be a different one?

---

## Referee Comment (RC2) · Maha Mdini (Referee) · 26 Nov 2020

The paper is generally well written and understandable. The text is fluent and precise. The paper introduces a modification of the Edit Distance used in the comparison of time series with extreme events. The technical content is sound and relevant. I would suggest the following comments:

1- In the shifting part of the cost function, replacing the linear function with a non linear function is very interesting. I would like to ask about the choice of the sigmoid? Have you tested other non linear functions and compared them (Tanh, RELU)?

2- The value of k is crucial for the behavior of the sigmoid and therefore for the cost function. Could you elaborate more on how you set the value of k. It would be

interesting to see the curve of DET as a function of k (like with $\tau$ in figure 8.C)

3- Line 222, I could not understand the definition of p. What is the difference between p and L?

4-Figure 12 and 13 have similar titles. Maybe, the overlap in figure 13 is 9 months.

5-It would be interesting to plot the curve of DET as a function of $\tau$ for flood events in figures 12 and 13 (like in figures 8,9 and 10 for synthetic data using Poisson process)

6- The size of figures 11 and 13 could be reduced.

7- In figure 15, the mean of the surrogates is around 0.15. This value may depend on the shuffling process used in the generation of the surrogates from the original data. Could you describe more how you randomized the original set (uniform/Gaussian process)?

---

## Author Comment (AC1) · 2 Feb 2021

With this review response letter, we will submit the revisions to the manuscript (MS# npg-2020-41) entitled "Recurrence analysis of extreme event like data" for publication in *Nonlinear Processes in Geophysics*. We would like to thank the referee for careful review and constructive suggestions which, we believe, have refined our manuscript. The responses to all the comments and suggestions put forth by the reviewers are enlisted below.

**Comment 1: In the shifting part of the cost function, replacing the linear function with a non linear function is very interesting. I would like to ask about the choice of the sigmoid? Have you tested other non linear functions and compared them (Tanh, RELU)?**

**Reply 1:** Thanks for the suggestion. The implementation of the logistic function is easy and we can use the idea of the *delay*, which has physical relevance in climate phenomena. The logistic function satisfies the triangular inequality criteria, proposed by Victor and Purpura.

The tanh function is a rescaled logistic function and therefore, comparing their results would be trivial.

The ReLU provides a linear cost function shifting, very similar to the original one of ED. The difference is an additional shift of the onset of the linear increase of the costs and, by this, introducing a zero cost for small shifts.

Nevertheless, we implemented the ReLU as a cost function according to our problem as follows:

$$f(t) = \begin{cases} 0 & \Delta t \leq \tau \\ \Lambda_0 \Delta t & \Delta t > \tau \end{cases} \tag{1}$$

where $\Lambda_0$ is computed using Eq. (2b) in the original manuscript and $\Delta t$ is the gap between two events. The cost will increase with the increment of the distance between events (Fig. 1). In our work, we assign the cost for deletion/insertion to be 1. So, the cost shifting will be optimized by ReLU function below the maximum cost 1.

Please note that strictly speaking, the ReLU function implemented in this way does not become a metric because two events with a small difference can be shifted without any cost.

[Figure]

Figure 1: ReLU function for different $\tau$ value

We used the ReLU cost function for the recurrence analysis of the Poisson process, repeating Poisson process, and the periodical Poisson process. The recurrence plots for all the above cases are shown in the Fig. 2.

In case of ReLU cost function, the DET value is smaller (Fig. 2) than DET of the logistic function (Fig. 8a, 9a, and 10a in the original manuscript). We also compute the DET for different

value of $\tau$ (Fig. 3). From Fig. 3 we see that it is bit difficult to find a range of $\tau$ values where the variation in DET is minimum.

[Figure]

Figure 2: RP of (a) Poisson process, (b) repeating Poisson process, and (c) periodical Poisson process using ReLU function for optimizing the cost of shifting.

[Figure]

Figure 3: DET comparison for 300 realizations of (a) Poisson process, (b) repeating Poisson process, and (c) periodical Poisson process using ReLU function.

**Comment 2: The value of k is crucial for the behavior of the sigmoid and therefore for the cost function. Could you elaborate more on how you set the value of k. It would be interesting to see the curve of DET as a function of k (like with Figure in figure 8.C)**

**Reply 2:** The logistic function has two parameters, among which $k$ is the rate parameter. Although $k$ determines the slope of the sigmoid, it does not have much impact on the outcome of the RQA, in particular of the DET measure. To demonstrate this, we consider a Poisson process and a periodical Poisson process and compute DET for different $k$ values. For the Poisson process, we keep $\tau = 0$ as the DET value using mED is maximum at $\tau = 0$, and for the periodical Poisson process we choose

$\tau$ to be 0 (DET value using mED is lower than ED) and 10 (DET value using mED is higher than ED). We find that DET remains more or less constant for the entire range of values of $k$ (Fig. 4). Whether the DET for the mED is higher or lower than ED is mostly determined by $\tau$, which can be interpreted as a delay between events and, thus, has a well understandable physical relevance for the considered system (and can be selected accordingly).

[Figure]

Figure 4: Comparison of DET for (a) Poisson process and (b) periodical Poisson process using mED for varying k and keep $\tau=0$, we use 300 realisations for each case.

**Comment 3: Line 222, I could not understand the definition of p. What is the difference between p and L?**

**Reply 3:** We modify the manuscript as follows:

"Each window consists of $n$ ($n \in \mathbb{N}, n > 1$) data points, so the window size is $w = n\Delta T$. For overlapping windows, a fraction of the data is shared between consecutive windows, denoted by $L \in \{1, 2, 3, .., n-1\}$ as the number of data points within the overlapping range, where $L = \varnothing$ signifies the non-overlapping case. The overlapping range is $O_L = L\Delta T$ in terms of percentage $O_L(\%) = \frac{L}{n}$."

**Comment 4: Figure 12 and 13 have similar titles. Maybe, the overlap in figure 13 is 9 months.**

**Reply 4:** We apologize for the typo. We will modify in the manuscript.

**Comment 5: It would be interesting to plot the curve of DET as a function of figure for flood events in figures 12 and 13 (like in figures 8,9 and 10 for synthetic data using Poisson process)**

**Reply 5:** Thank you for the comment. We will include the following discussion and figure in our manuscript:

"In order to compare *Edit distance*(ED) and the *modified Edit-distance*(mED) in case of flood events, we compute the RP and the *determinism* using both the methods (Fig. In case of mED, we vary the $\tau$ in the range 1 to 60 days. However, in case of ED we get the DET value for a particular

$\tau$. For mED, we can vary the $\tau$ and compute the DET for different time scale. From Fig. 5, we find higher DET value for certain delay value.

[Figure]

Figure 5: Comparison of DET for flood events using ED (horizontal line) and using mED (curved line) for 6 months overlapping (in blue color) and for 9 months overlapping (in red color)

**Comment 6: The size of figures 11 and 13 could be reduced.**

**Reply 6:** We will modify in the manuscript.

**Comment 7: In figure 15, the mean of the surrogates is around 0.15. This value may depend on the shuffling process used in the generation of the surrogates from the original data. Could describe more how did you randomize the original set (uniform/Gaussian process)?**

**Reply 7:** We exchange row (i) and column (j) simultaneously in the original recurrence plot with different pairs $(i, j)$ as randomly chosen through a uniform process. In this way, we create 50 surrogate recurrence plots. We will include this information into the manuscript.

---

## Author Comment (AC2) · 2 Feb 2021

With this review response letter, we will submit the revisions to the manuscript (MS# npg-2020-41) entitled "Recurrence analysis of extreme event like data" for publication in *Nonlinear Processes in Geophysics*. We would like to thank the referee for careful review and constructive suggestions which, we believe, have refined our manuscript. The responses to all the comments and suggestions put forth by the reviewers are enlisted below.

**Comment 1: "In the paragraph starting at line 75, I propose to write after the first sentence that (1) the aim is to determine a distance measure between each segment and (2) that the transformation is a means to an end, that is, the aim is creating a cost of the transformation as the distance measure."**

**Reply 1:** We modify the paragraph according to the suggestion as below:

"In order to apply the edit distance as a distance measure for, e.g., recurrence analysis, the whole time series is divided into small, possibly overlapping, segments (windows), which should contain some data points (Fig. 1). The aim is to determine a distance measure between every pair of segments. As a distance, we use the effort of transforming one segment into another using a certain set of operations. For this a combination of three elementary operations is required: (1) delete or (2) insert a data point, and (3) shifting a data point to a different point in time; each of these operations is assigned a cost. We obtain the distance measure by minimizing the total cost of this transformation."

**Comment 2: "Then, I could not find the term "edit distance" in the original paper from Victor and Purpura 1997. I think it would be good to make a remark for readers wanting to dive into the origins."**

**Reply 2:** We make the following modification to be more precise about the origin of the term:

"Event-like time series can be analyzed in their unaltered form by considering a time series of discrete events as being generated by a point process. Victor and Purpura (1997) presented a specific distance metric to calculate a distance between two spike trains (binary event sequences) as a measure of similarity. Hirata and Aihara (2009) extended this idea for analysing event-like time series and named it *edit-distance* [1]."
We will add Ref. [1] in our manuscript.

**Comment 3: "Line 254: did you really use eq. (4)? That is the Heaviside function, and you claim in chapter 3 that you use the logistic function."**

**Reply 3:** We apologize for the typo in line 254 of the original manuscript. The correct equation number should be Eq. (5), which is the logistic function. We modify the line accordingly as follows:

"For ED (Eq. 1), the cost parameter for shifting is calculated according to Eq. (2a) and $\Lambda_s = 1$, whereas Eq. (5) is used for mED, recurrence threshold $\varepsilon = 10$ percentile of the distance matrix for each case."

**Comment 4: "- Paragraph from line 296 on: I don't understand this completely. You apply a threshold on the data of daily discharge? You determine the percentile per year? A few events - you mean 3 to 4 (you wrote 3/4 = three quarters)?"**

**Reply 4:** We modify the paragraph from line 296 as below to make the language clearer and

more understandable:

"To find the independent events from the time series we follow the procedure below:

1. First, we select events above an arbitrary threshold, say $99^{th}$ percentile value of the daily discharge time series for a particular year, which gives about 3 to 4 events per year.

2. Next, if several successive days fall above the threshold forming a cluster of events, we pick only the day which has the maximum discharge value and remove the remaining events of the same cluster.

3. Then we lower the threshold by 0.1 percentile (the threshold is lowered from $99^{th}$ to $90^{th}$) and repeat the same procedure as above until we get the desired number of independent events.

**Comment 5:** "- Line 303 recurrence threshold: this parameter is quite crucial in RQA. You set it to 8% of the distance distribution. Is there any specific reason for this value (in the Poisson events you used 10%), was there a reasoning or trials to get to this value? Then please include these in the paper. If it was "just chosen", then write this as well please.?"

**Reply 5:** We agree that the choice of recurrence threshold is very crucial for obtaining the recurrence plot and in recurrence quantification analysis as well. The selection of the threshold based on a certain percentile of the distance distribution makes the recurrence quantification more stable [2]. Keeping in mind that this threshold should neither be too high nor too low, we chose it as 8 to 10% of the distance distribution based on a number of trials. For each example in the manuscript, we chose that percentile of the distance distribution which retains the stability of the recurrence quantification while giving the most prominent pattern on the recurrence plot. We will modify the manuscript by adding Ref. [2] and the justification of the choice of the recurrence threshold as given above and we will set the recurrence threshold as 10% of the distance matrix for all the cases.

**Comment 6:** "I am missing a discussion of the difference of your modified edit distance and the original one on the Mississippi flood events, i.e.,comparing figure 12 and 13 with figure 14. You mention that you computed standard edit distance and figure 14 in line 304, but I am missing a discussion about the outcome of this, or the comparison with your modified distance."

**Reply 6:** Thank you for the comment. We will include the following discussion and figure in our manuscript:

"In order to compare the *edit distance* (ED) with the *modified edit distance* (mED) in the study of flood events, we compute the RP and *determinism* DET using both methods. For ED, there is no scope to implement a delay, we get the DET value for a constant temporal gap. In case of mED, because of the predefined delay $\tau$ we can study the behaviour at different time scales. In order to do so, we vary the parameter $\tau$ in the range from 1 to 60 days. For mED the DET values is slightly lower than ED up to $\tau = 30$ days (Fig. 1). But after that, in the range of 30 to 60 days of $\tau$, DET becomes higher for mED. So, for this particular time series we get more deterministic behaviour for delay in range 30 to 60 days.

**Comment 7:** "- A final remark: what do you think about modifying the title or amending it by "...using a modified distance measure"

[Figure]

Figure 1: Comparison of DET for flood events using ED (horizontal line) and using mED (curved line) for 6 months overlapping (in blue color) and for 9 months overlapping (in red color)

**Reply 7:** Thanks for the suggestion. We can modify the title as follows:

"Modified distance measure to analyze event like time series."

**Technical corrections**

**Comment 8:"- Line 60/61: I suggest writing either "A flood time series shows..." or "Flood time series show...""**

**Reply 8:** We will modify the line in our manuscript as:

"We demonstrate the efficacy of the proposed modified edit distance measure by employing recurrence plots and their quantification for characterizing the dynamics of flood time series from the Mississippi river in the United States. A flood time series shows a complex time-varying behaviour"

**Comment 9: "- Line 67: To be strict, "RP" is the abbreviation of "recurrence plot"; "recurrence plot analysis" (RPA) is also sometimes used, but "recurrence quantification analysis" (RQA) is much more common."**

**Reply 9:** Throughout our manuscript we strictly follow either of the conventional abbreviation RP or RQA corresponding to recurrence plot and recurrence quantification analysis respectively. We apologize that we overlooked the glitch in line 67. We modify the line as follows:

"Distance measurements between two data points play an important role for many time series analysis methods, for example, in recurrence quantification analysis (RQA) (Marwan et al., 2007), estimation of the maximum Lyapunov exponent (Rosenstein et al., 1993),"

**Comment 10: "Line 82: "where alpha and beta... at times ta(alpha)" - isn't there "and tb(beta")missing?"**

**Reply 10:** There should be a $t_b(\beta)$ in that line. We include the following correction in the manuscript:

"where $\alpha$ and $\beta$ are events in segments $S_a$ and $S_b$ occurring at times $t_a(\alpha)$ and $t_b(\beta)$;"

**Comment 11:** "The definition of determinism DET has been done with a minimum line length of 2, as one can see in eq. (8). Please consider to write down "with a minimum line length of 2" or similar to make this clear"

**Reply 11:** We modify the paragraph in the following way:

"One of the most important measures of RQA is the determinism (DET), based on the diagonal line structures in the RP. The diagonal lines indicate those time periods where two branches of the phase space trajectory evolve parallel to each other in the phase space. The frequency distribution $P(l)$ of the lengths of the diagonal lines is directly connected to the dynamics of the system (Marwan et al., 2007).

$$DET = \frac{\sum_{l=l_{\min}}^{N} lP(l)}{\sum_{l=1}^{N} lP(l)}.$$

In our study, we choose $l_{\min} = 2$."

**Comment 12:** "Figure 6 caption "$\Delta t$ is the gap between two events in kept up to the maximum value 6." - I do not understand this sentence."

**Reply 12:** We will modify the caption the manuscript in the following way:

"Variation of the cost contributed by shifting an event by $\Delta t$ for different parameter values $\tau = 2, 3$, and 4.7 using (a) the logistic function with $k = 1$ and (b) the Heaviside function. "

**Comment 13 and 14:** "Line 221: The symbol OL for the overlapping range might be confusing, especially in an equation in which on the right hand side (LdeltaT) L and deltaT are multiplied (I was wondering in the first place whether O shall be multiplied by L on the left hand side). Consider using an index $O_L$ (sorry for not being able to display it here)"

"Line 222: I do not understand the last sentence: the number of shared data points is L, is it not? This can be expressed of course as OL/deltaT. omega times deltaT minus OL makes no sense anyway, because the first summand has unit time squared and the second summand has unit time."

**Reply 13 and 14:** We modify the manuscript as follows:

"Each window consists of $n$ ($n \in \mathbb{N}, n > 1$) data points, so the window size is $w = n\Delta T$. For overlapping windows, a fraction of the data is shared between consecutive windows, with the number $L \in \{1, 2, 3, .., n-1\}$ of data points within the overlapping range, where $L = \varnothing$ signifies the non-overlapping case. The overlapping range is $O_L = L\Delta T$ and in terms of percentage $O_L(\%) = \frac{L}{n}100\%$."

**Comment 15:** "Line 223: I suggest removing the "and"."

**Reply 15:** We will remove it in the manuscript.

**Comment 16:** "Line 234: "...different signal"s - "s" is missing"

**Reply 16:** We apologize for the typo. We will correct it in the manuscript.

**Comment 17:** "Line 306 a dot after "Fig" is missing"

**Reply 17:** We apologize for this typo in line 306 of the original manuscript. We will insert the dot in the manuscript.

**Comment 18: "Line 306 and 307: you wrote Fig. 13d twice. I guess the first shall be a different one?"**

**Reply 18:** We apologize for this error. In line 306, we will replace the reference to Fig. 13d with the correct figure number Fig. 13c.

**References**

[1] Yoshito Hirata and Kazuyuki Aihara. Representing spike trains using constant sampling intervals. *Journal of Neuroscience Methods*, 183(2):277 – 286, 2009.

[2] K. Hauke Kraemer, Reik V. Donner, Jobst Heitzig, and Norbert Marwan. Recurrence threshold selection for obtaining robust recurrence characteristics in different embedding dimensions. *Chaos: An Interdisciplinary Journal of Nonlinear Science*, 28(8):085720, 2018.